# Digital measurement of ocular microtremor in Parkinson's disease: Protocol for a pilot study to assess reliability and clinical validation

Lisa Graham[1,2], Rodrigo Vitorio[1], Richard Walker[3], Alan Godfrey[4], Rosie Morris[1,3], Samuel Stuart[1,3,5]*

1 Department of Sport, Exercise and Rehabilitation, Northumbria University, Newcastle upon Tyne, United Kingdom, 2 Gateshead Health NHS foundation trust, Gateshead, United Kingdom, 3 Northumbria Healthcare NHS foundation trust, North Shields, United Kingdom, 4 Department of Computer and Information Sciences, Northumbria University, Newcastle upon Tyne, United Kingdom, 5 Department of Neurology, Oregon Health and Science University, Portland, Oregon, United States of America

* samuel2.stuart@northumbria.ac.uk

**Data Availability Statement:** No datasets were generated or analysed during the current study. All relevant data from this study will be made available upon study completion.

## Abstract

Ocular microtremor (OMT) is a fixational eye movement that cannot be seen with the naked eye but is always present, even when the eye appears motionless/still. The link between OMT and brain function provides a strong rationale for investigation as there lies potential for its use as a biomarker in populations with neurological impairments. OMT frequency is typically 70-80Hz in healthy adults and research suggests that this will be reduced in those with neurological disease such as Parkinson's Disease (PD). This study aims to examine OMT in people with PD compared to healthy older adults. This is an exploratory, observational study that will use a novel handheld device–The iTremor ONE, which has been developed to rapidly, non-invasively assess and evaluate OMT frequency. This device uses incident laser technology directed at the sclera. People with PD who meet the inclusion criteria will participate in a home-based assessment involving cognitive, motor (using the UPDRS-III) and OMT measures. With OMT as the primary outcome, assessment with the iTremor is quick, taking just three seconds to obtain a reading. People with PD will be invited into the laboratory to perform extensive cognitive assessments along with an assessment of balance, gait, and turning using wearable sensors. People with PD will be assessed both off, and on, their anti-parkinsonian medication following a 12 hour washout period. We will recruit 30 People with PD, 30 people with suspected PD and 30 age-matched healthy control participants for assessment of OMT. 20 People with PD will complete a test-retest reliability assessment at the same approximate time, exactly one week after their initial visit under the same conditions to explore consistency. This will be the first study of its kind to non-invasively investigate OMT frequency as a marker/monitor for PD with advanced technology that could be used within the clinic, laboratory, or home. Identifying OMT as a PD biomarker could better support clinical assessment, enabling improved provision of care to patients with advanced disease monitoring.

**Clinical trial registration:** This trial is registered at clinicaltrials.gov (NCT06051877; September 2023).

**Funding:** This study is based at the Physiotherapy Innovation Laboratory (Website: www.pi-lab.co.uk, Twitter: @Physio_In_Lab) and has been funded by a Northumbria University PhD studentship in collaboration with Head Diagnostics Ltd. (Dublin, Ireland). Dr Samuel Stuart is supported, in part, by a Parkinson's Foundation Post-doctoral Fellowship for Basic Scientists (PF-FBS-1898-18-21) and a Clinical Research Award (PF-CRA-2073). Private Physiotherapy Educational Foundation funding also supported this research (PPEF – #368; PI: Stuart).

**Competing interests:** The authors have declared that no competing interests exist.

# Introduction

Parkinson's disease (PD) is the second most common neurodegenerative disease and the world's fastest-growing neurological condition [1], for which there are limited non-invasive biomarkers. PD is highly prevalent, with the mean age of onset being just 57 years-old and affecting ~2% of the world's population over age of 60 [2]. PD is a progressive condition for which there is no known cause [3]. With the life expectancy rising globally, an increasing number of people are being diagnosed with PD and people with PD are living with their symptoms for longer. With this in mind, there is increasing pressure on healthcare services due to the progressive nature of the condition. Early intervention and monitoring of treatment response and disease progression through identification of biomarkers is essential [4]. The current method of diagnosis in PD is dependent on expert clinical judgement and there is a large subjective component. Dopamine transporter imaging (DaTSCAN) can also be used to confirm the diagnosis. DaTSCAN is an in-vivo, nuclear imaging technique which uses an IoflupaneI-123 injection to visually assess dopamine transporter density in the brain [5]. This, however, is invasive, expensive, time consuming and usually unpleasant for the patient. It requires a trained operator and can mean patients are put on a waiting list to get an appointment in addition to the initial wait to see the consultant.

Eye movements are well understood and provide an important insight to underlying neuropathology [6]. Eye movements have been identified as a likely biomarker for PD but there is no consistent, robust, and reliable gold standard method to objectively quantify these [7, 8]. Eye movement can indicate basal ganglia function [9], which is specifically relevant to PD due to the loss of dopamine in the substantia nigra. Anomalies are more evident in voluntary eye movements in the initial stages of PD whereas visually guided (reflexive, involuntary) saccades may also be involved in later stages [18] and deficits persist throughout disease progression [10]. In PD deterioration of dopaminergic mediated pathways in the BG causes overactive superior colliculus (SC) inhibition which in turn, prevents the SC from triggering the brainstem to generate saccades [11]. For example, pro-saccade performance (directing gaze towards a target) has been identified in predicting decline in PD, highlighting the potential benefits of accurate eye tracking in detecting these [2]. The difficulty of using eye movements as a marker of PD largely relates to the need for voluntary control of tasks, such as voluntarily moving the eyes towards or away from targets, which can become challenging due to the need to understand instructions. Additionally, there is much variation in the literature regarding the effects of dopaminergic medication on eye movements. This was illustrated in a study using a 12 hour 'OFF' period, comparing people with PD when they were 'ON' and 'OFF' their dopaminergic medication (in this case levodopa) which found that while there was high inter-subject variability, dopaminergic medication prolonged saccadic latency [7]. In other research, again using a 12 hour 'OFF' period, similar effects were reported showing that levodopa slowed reflexive pro-saccades (directing gaze towards a target), but the authors also reported improved voluntary anti-saccades (directing gaze in the opposite direction of target) [12]. In a more recent study looking at saccadic response to levodopa, participants refrained from taking their morning dose of anti-Parkinson's medication [13]. They were tested upon arrival (OFF), following which participants then took their medications, waited 60 minutes, and were tested again (ON). Here authors reported that pro-saccadic latency was prolonged by dopaminergic medication in PD but that no other statistically significant change was observed in any other saccadic parameters [13]. Consequently, involuntary eye movements may provide a useful biomarker for PD that is responsive to dopaminergic medication. Testing using ON and OFF periods with dopaminergic medication has been extensively performed in research with no detrimental effects to the individual. The existing evidence for eye movement response to

dopaminergic medication therefore provides rationale for investigating involuntary eye movements in an ON and OFF state utilising a 12 hour 'OFF' period that has previously been described in similar eye movement research.

Ocular micro tremor (OMT), first described in 1934, is an involuntary eye movement that is present even when the eye is apparently still [14]. OMT is considered a fixational movement and is a constant small amplitude, high frequency tremor of both eyes which goes unnoticed by the individual and cannot be seen by the naked eye [15]. OMT has been linked to the constant activity of the extra-ocular muscles stimulated by impulses from oculomotor neurons found in the brainstem [15–19] and thought to be related to tonic neural discharges. Clinical use of OMT has been explored in cases of PD [20], Multiple sclerosis [21], brainstem death [17] and, measuring depth of anaesthesia [22, 23]. OMT frequency is historically measured using the piezoelectric method [24]. In those with neurological disease and impairment, OMT frequency is reported to be reduced compared to healthy controls [17, 20, 21, 25]. OMT frequency is also known to be affected by alcohol [26], caffeine [27] and age [28]. Whereas, time of testing [26] and gender [29] have not been found to affect OMT and no differences have been found between left and right eye readings [30]. When reviewing findings on OMT, a study described the frequency of OMT to range from 70 – 130Hz in healthy individuals [29]. However, it is suggested that discrepancies in the estimated normal frequency of OMT oscillations seen in the literature are likely due to lack of an efficient gold-standard technique for measurement meaning methods differ between studies [24, 31]. Previous work has illustrated that OMT frequency declines with age, with the authors indicating normal OMT values are different for those over the age of 60 years [28].

Neurological disease also significantly decreases OMT frequency when compared to healthy controls. For example in PD, a study reported a mean OMT frequency of 67.68Hz (Healthy range = 70 – 130Hz [29]). In another study focusing on OMT and idiopathic PD, authors observed two groups–an ON group and OFF group [20]. These were determined using the bradykinesia subsection of the Webster Scale [32]. A score of 0 or 1 was considered ON (n = 12), and a score of 2 or 3 was considered OFF (n = 10). The healthy controls in this study had a mean OMT frequency of 81.64Hz (SD = 6.10) and both PD ON, and PD OFF, groups exhibited significantly lower OMT frequencies of 73.78Hz (SD = 5.55) and 58.88Hz (SD = 10.35) respectively; the mean frequency of the OFF group was significantly lower than that of the ON group. This preliminary evidence suggests that People with PD have reduced OMT, thus supporting the need for more robust evidence.

PD related motor symptoms tend to only be noticed when there has been a ~80% depletion of striatal dopamine and so identification of the disease using OMT as an objective measure could benefit prognosis and likely help identify subtle changes in the early stages [33]. Observing OMT through multiple recordings in patients in an ON and OFF state could yield insight to individual responsiveness to dopaminergic medication. Exploring OMT as a biomarker for PD using accessible novel technology could facilitate an objective diagnosis and monitoring of the disease and it's progression [24]. Overall, this presented protocol aims to examine OMT as a potential marker for PD and for disease monitoring. To achieve this, the following aims are outlined:

Analytical Validation: (1) Investigate the test-retest reliability of the iTremor ONE device in measuring OMT frequency (Hz) in People with PD; (2) Investigate the feasibility of measuring OMT using a hand-held device in a real-world clinical setting. This will cover the reliability and feasibility of OMT measurement in People with PD.

Clinical Validation: (1) Compare OMT frequency in People with PD and age-matched controls, and suspected PD patients; (2) Explore relationships between OMT measurement in People with PD and other clinically relevant outcomes (e.g., UPDRS); and (3) Examine the

effect on OMT of dopaminergic medication in PD. This will cover known groups validity, convergent/divergent validity and response to a known intervention in People with PD.

## Methodology

### Design

This observational study will involve three steps; Step 1) Recruitment, Step 2) Laboratory assessment of People with suspected PD, confirmed PD and healthy controls, Step 3a) home assessment of people with confirmed PD, and step 3b) re-test home assessment of people with confirmed PD. Participants do not have to be involved in all steps of the study to be recruited (i.e., they could do only the lab or the home assessment, or both). * in 'OFF' and 'ON' states.

### Study setting

Due to the non-invasive nature and short reading time required for the OMT measurement, data can be collected at the Clinical Gait Laboratory in Northumbria University, and within the participant's home. For data collection for ON/OFF medication states, data will be collected in the participants' home. This minimises the time for which the participant needs to be OFF their medication and reduces risks associated with travelling to the lab in their OFF state.

### Ethical approval and trial registration

A Northumbria University research ethics committee granted ethical approval (Project No. 0034). The trial has been reviewed and registered with Clinicaltrials.gov (NCT06051877). This study also received NHS REC approval (REF: 23/WM/0004), and MHRA letter of no objection (REF: CI/2023/0031/GB). Recruitment began 10th of June 2023.

### Participants

**Step 1) Recruitment.** Participants will be a volunteer sample recruited from Northumbria Healthcare NHS foundation Trust. Participants will be seen in clinic by a Movements Disorders specialist at these sites where people are referred for suspected PD diagnosis, and those with diagnosed PD will be identified. Potential participants will be invited to consider the study and referred to the investigators who will attend the clinics. Healthy controls could include spouses/ siblings/ other volunteer older adults. We will recruit healthy older adults from advertisement using posters, which will be placed within neurology and geriatric departments. We will also recruit older adult and PD participants from previous or ongoing studies at Northumbria University where the participant has given written informed consent to be contacted regarding future research studies, and from local PD support groups.

*Inclusion / Exclusion criteria*. PD participants will be included if they have a diagnosis of PD according to UK brain bank criteria. PD and older adult participants will be excluded if there is any psychiatric co-morbidity, a history or evidence of head injury or ocular disease (such as cataracts), they have a clinical diagnosis of dementia or other severe cognitive impairment (measured using the MoCA with a cut off score of <21). For those involved in Step 3 of the project, PD participants must be consistently taking dopaminergic ('anti-Parkinson's') medication.

**Step 2) Lab assessment.** We will recruit 30 confirmed PD, 30 suspected PD, and 30 age-matched healthy control participants for laboratory and home-based assessment of OMT (Fig 1). As this is a pilot study of a novel OMT measurement device and we are interested in the feasibility of measuring OMT in a real-world clinic setting rather than determining efficacy of OMT for differential diagnostics in PD within clinical settings, a formal sample size

| STUDY PERIOD | | | | | |
|---|---|---|---|---|---|
| | Enrolment | Allocation | Post-allocation | | |
| TIMEPOINT** | *-t₁* | 0 | *t₁* (Lab Assessment) | *t₂* (Home Visit 1) | *t₃* (Home Visit 2) |
| **ENROLMENT:** | | | | | |
| **Eligibility screen** | X | | | | |
| **Informed consent** | X | | | | |
| **Allocation** | | X | | | |
| **INTERVENTIONS:** | | | | | |
| ***N/A*** | | | | | |
| **ASSESSMENTS:** | | | | | |
| **Cognitive Assessments** | | | | | |
| *Montreal Cognitive Assessment (MoCA)* | | | X | X | X |
| *Benton's Judgment of Line Orientation (JLO)* | | | X | | |
| *Forward Digit Span* | | | X | | |
| *Trail Making Tasks A&B (TMT A/B* | | | X | | |
| *CLOX 1&2* | | | X | | |
| *Visual assessments* | | | | | |
| *LogMar* | | | X | | |
| *LogCS* | | | X | | |
| *Motor Assessments* | | | | | |
| *n-FOG* | | | X | | |
| *FES-I* | | | X | | |
| *MDS UPDRS-III* | | | X | X | X |
| *H&Y* | | | X | X | X |
| *Gait & Balance* | | | X | | |
| *OMT Assessment* | | | | | |
| *iTremor* | | | X | X | X |

**Fig 1. SPIRIT diagram, schedule of enrolment and assessments.**

calculation was not conducted. We have based the sample size on previous studies of eye-tracking in PD that have used samples <30 per group and found significant differences compared to controls [34].

**Step 3a) Home assessment 1.** We will recruit 30 people with confirmed PD to participate in a home based assessment.

**Step 3b) Home assessment 2.** We will recruit 20 people with confirmed PD who participated in step 3a to participate in a second re-test home based assessment of reliability.

## Equipment

**iTremor (see Fig 2).** A portable, handheld device that uses optical sensors to measure OMT, a method based on measuring angular displacement using laser speckle correlation of images recorded in the Fourier plane of a lens [35]. This method is able to detect OMT

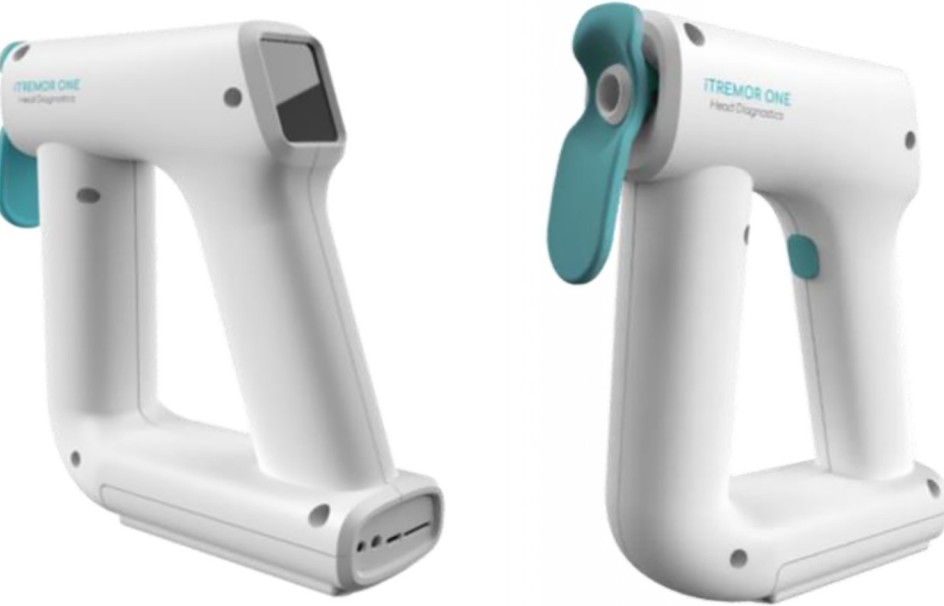

**Fig 2. iTremor ONE device.**

frequency regardless of other eye movements, head movements and tear flow [36]. The iTremor device captures 500 frames per second, at a resolution of 2 mrad and a dynamic range of 2–9.52 mrad. Recording of OMT using the iTremor device takes ~3 seconds and is typically conducted in a seated position with eyes in the neutral position.

**Inertial sensors.** Portable, light-weight inertial measurement units (Opals, APDM, USA) will be placed on the participant at various locations (i.e., feet, lower back, chest, wrists) using elasticated Velcro straps. These sensors contain accelerometer and gyroscope technology, have a high sampling rate and are small in size (~44mm x 50mm x 14mm (LxWxH)). The raw data collected from these is stored on the sensors and wirelessly streamed to APDM Mobility Lab, to provide automatic output of gait and balance outcomes. These outcomes include stride length, step time variability, gait speed, and trunk and arm range of motion. Turn characteristics are also recorded during walking.

**Testing protocol.** A schedule of enrolment and assessments is presented in Fig 1.

**Step 1: Clinical recruitment.** A member of the research team will provide the participant with the necessary oral and written explanations to obtain their signed/written informed consent before beginning the study. Participants will be invited to the clinical gait laboratory at Northumbria University, Coach Lane Campus for the lab-based assessment.

*Diagnosis outcome.* Researchers and clinicians involved in the study will make note of the diagnostic outcome for those with suspected PD (i.e., either diagnosed with PD or not) provided by either Clinical or DATSCAN outcome, which will be obtained from the participant medical record and added to the secure participants database (i.e., PD; YES or NO).

**Step 2: Laboratory assessment (Suspected PD, confirmed PD and HC groups).** *Demographics.* The participant will then be interviewed by a member of the study team. The interview will include questions regarding education level, falls history, current medications, history of head injury, activity level, and side dominance as well as basic demographics such as age and gender. We will ask participants for their medications currently used (so that we can calculate the levodopa equivalent daily dosage (LEDD) [37, 38].

*Cognitive assessment*: Participants will also undergo a battery of cognitive, motor, and visual assessments. The cognitive assessments should last around 30 minutes and consist of a series of pen and paper tasks which the participants will be guided through by a member of the research team. Global cognition is assessed using the Montreal Cognitive Assessment (MoCA) [39]. Benton's Judgement of Line Orientation test assesses visuospatial skill (JLO) [40]. The trail making test of visual attention (TMT A and B) is used to assess executive functioning [41] as well as Royall's Clock Drawing task (CLOX 1&2) [42]. Forward digit span [43] is used as a measure of working memory. Participants will also complete a series of computerised tests to assess attention using a button pressing task. This attention battery is a valid means of testing attention and has been implemented in previous studies involving People with PD [44].

*Visual assessment*: The visual assessments are conducted binocularly before using an eye occluder to measure for the left and right eyes independently. The visual assessments should take 10 minutes to complete.

*Visual acuity—LogMAR*. This test is a simple visual test to assess participants ability to read at a distance. The Logarithm of the Minimal Angle of Resolution (LogMAR) is used as a standardised binocular measure. Participants are seated at a 4-meter distance from the chart and asked to read the letters aloud from left to right starting at the top left. The participant is stopped if two incorrect letters are given. LogMAR is calculated using the following formula: LogMAR = (score of the line before termination)–(0.02 x the number of errors) + (0.02 x correct answers before terminal line). A LogMAR score of 0.00 is equivalent to 6/6 (20/20) vision. The LogMAR charts are scored by letter instead of row making them a more precise measure of visual acuity.

*Contrast sensitivity—LogCS*. This is measured using the Mars CS sheets. These sheets consist of 48 Latin letters of uniform height, and the contrast from the white background decreases with each letter. The participant is seated 50 centimetres (cm) from the sheet and again reads from left to right, starting at the top left. Errors are recorded by the investigator and the test is terminated when two consecutive errors are made. LogCS is calculated using the following formula: (value of final correct letter before stopping)–(number of errors prior to stopping x 0.04). Again, answers are recorded for binocular vision then for left, and right eyes independently using an occluder.

*Motor assessment*: To assess disease severity and motor symptoms in People with PD, the Movement Disorder Society Unified Parkinson's disease Rating scale part III (UPDRS-III) is used [45]. The motor assessments should take around 30 minutes. The UPDRS will be conducted first before setting up the sensors for the balance and gait assessments. People with PD will be categorised based on their disease severity using the Hoehn and Yahr scale (H&Y) [46]. This is a widely used clinical scale for defining motor symptoms in PD, which is a part of the UPDRS-III. It consists of five stages–stage I, unilateral symptoms with no functional disability, to stage V, complete disability and patient is bedbound. This scale is easily applicable and widely used to assess PD. There will also be a short self-report questionnaire on freezing of gait, using the new Freezing of Gait Questionnaire (nFOGQ) [47]. In addition, the Falls Efficacy Scale–International (FES-I) [48], a self-report evaluation to assess fear of falling will be completed.

*Balance*. Participants will stand for several periods of up to 2 minutes under different conditions (i.e., eyes open, eyes closed, and on a foam block with eyes open, and closed) with balance measured via mobile inertial measurement units (APDM, USA). These sensors are located at the wrists, feet, chest, and lower back to measure postural sway.

*Gait*. Participants will walk back and forth over a marked out 10m distance for several periods of up to 2 minutes under different conditions (i.e., normal walking and walking while

repeating numbers) with gait measured via mobile inertial measurement units (APDM, USA). Turns are included in this walking sequence. These sensors are located at the wrists, feet, chest, and lower back.

*OMT assessment*: OMT will be measured using the iTremor Device (Fig 2). The participant will be asked to sit comfortably with their head resting on the chair's headrest. If the participant wears glasses or contact lenses they will be asked to remove these for the duration of the recordings. The participant is asked to look straight ahead at a target on the wall without straining their eyes for a ~3 second period. The participant remains relaxed whilst the researcher positions the device, gently leaning on the participants cheek and brow bone. The guide light is lined up from the side of the eye with the scleral surface of the eye and the researcher cues the participant to not blink before pressing the trigger on the device. An optical beam is emitted from a diode in the device at low power with light scattered back from the eye and reflected back into the device and passed to a video imaging sensor. Three readings will be taken for each eye. If the participant blinks, the reading is just repeated. Unwanted eye movements or blinks are filtered out in raw data post-processing. A frequency will be shown on the device's display screen in Hz. This is repeated three times for each eye independently.

**Step 3a: Home-based assessment (Confirmed PD only).**   Within ~1 month of Step 2, participants will be asked to abstain from their dopaminergic medication overnight for 12 hours, and researchers will attend their home in the morning prior to their morning dose, to conduct the home-based testing.

*Conducted in OFF medication state*. We will collect the following details when OFF medication: MoCA and UPDRS-III.

*Conducted in OFF and ON medication states. OMT*: OMT will be measured using the iTremor Device in the same manner as Step 2 (see above for details). OMT testing will be conducted immediately when attending the home (OFF state, after 12hours medication withdrawal), and then again 60 minutes after the participant has taken their usual dopaminergic medications.

**Step 3b: Home-based assessment 2 (Confirmed PD only).**   Participants from step 3a will be invited to participate in an additional home based assessment to explore the reliability of OMT measurement in PD using the iTremor ONE device. For this, we then visit the participant at the same time as step 3a, one week later. To perform the test re-test reliability, we ask participants to take their medication at the same time as they took it for step 3a. We will collect the following details when ON medication: MoCA, UPDRS-III and OMT (60 minutes after the participant has taken their usual dopaminergic medication).

**Safety considerations.**   There are no major disadvantages to participation in this study and no adverse events are anticipated. All measurements are non-invasive and quick meaning the participant is at no/minimal risk, and tasks performed are the same as participants would perform at home/in the community (i.e., no more risk than usual activities). The iTremor device uses optical technology to direct light across the scleral surface (not directly into the eye/pupil) at a safe frequency for use with human eyes, which has been verified by a consultant optical safety expert. Anyone who operates the iTremor device will be trained to do so. The component of the iTremor device that comes into contact with the participants skin is made of silicone and therefore should not cause any harm.

All equipment used with participants will be wiped using iso-propyl 70% alcohol cleaning wipes between each participant to ensure good hygiene. All research equipment (iTremor, inertial sensors etc.) provides no risk of electric shock from electrical failure as the devices are battery powered and therefore low voltage.

## Data analysis

**Primary outcome.**   The primary outcome for the study will be frequency (Hz) of OMT, measured via the handheld iTremor device (Head Diagnostics Ltd, Dublin, Ireland). Raw data will be analysed using an algorithm built into the handheld device. Raw data will be securely stored on a removable SD card within the device, which will only be accessed by the investigators.

**Secondary outcomes.**   Secondary outcomes for the study include cognition, visual function, motor symptoms (UPDRS-III), gait and balance. Diagnosis will also be recorded from the medical records for Step 1; specifically positive or negative PD diagnosis (i.e., PD; YES or NO).

Within clinic feasibility outcomes will include the following:

The number of adverse events (mild, moderate or severe) during the OMT measurement, if any, are recorded in an incident log. The total number will be assessed and compared.

The number of OMT measurement device issues (e.g., breakdowns, inability to record OMT etc.) and success of data recording (percentage of failed recordings / missing data). This may be device related or due to other factors such as environmental or patient specific difficulties.

Compliance adherence in terms of the number of completed OMT recordings from patients and their timeliness in terms of collection in the patient visit window.

**Data processing.**   The study will comply with the General Data Protection Regulation (GDPR) and Data Protection Act 2018, which require data to be de-identified as soon as it is practical to do so. The processing of the personal data of participants will be minimised by making use of a unique participant study number only on all study documents and any electronic database(s). All data samples collected as part of this study will be anonymised with participants being assigned a unique study number (e.g. iTrem_PD_01, iTrem_PD_02 etc.). All data are entered into an electronic database using unique study codes for each participant and are securely stored on a password-protected computer database. We will keep one hard copy of the assessment in locked filing cabinets in the Clinical Gait Laboratory, Coach Lane, Northumbria University. This is the only place where we store any personal details like names and addresses. This information is kept locked away and is only available to people directly running the study. These people will treat the information in the strictest confidence.

**Statistical analysis.**   Statistical analysis will be undertaken using SPSS version 26 or more recent versions (SPPS, Inc. an IBM company). All statistical tests will be carried out at the 5% two-sided level of significance. Demographic characteristics and baseline data will be summarized using descriptive statistics, including means, standard deviations, median, minimum, maximum and inter-quartile ranges for continuous or ordinal data and percentages for categorical data. The descriptive statistics will be tabulated and presented graphically for clarity.

*Analytical validation aim I. Reliability*: To assess test, re-test reliability in PD, OMT frequency outcomes from the iTremor One device will be assessed on two separate home-based sessions and compared. Absolute agreement between sessions will be assessed using intra-class correlations (ICC2,1). ICCs will be interpreted as; poor <0.5, moderate 0.50–0.75, good 0.75–0.90 and excellent >0.90 [49]. Additionally, Bland-Altman graphs, with mean differences and limits of agreement will compare OMT between the two sessions, and Pearson's correlations will also be used.

*Analytical validation aim II. Feasibility*: Feasibility outcomes will also be descriptively reported (e.g., number, percentages etc.) in relation to measurement of the OMT with the iTremor.

*Clinical validation aim I. Known groups validity*: To examine whether OMT can differentiate confirmed PD from suspected PD participants, we will use an analysis of variance (ANOVA), with group (confirmed PD vs suspected PD) as a between-subject factor. Although the study is not envisioned or powered to determine the efficacy of the OMT measurement for differential diagnosis of PD, we will further explore the difference in OMT between those with a positive diagnosis and those that did not receive a PD diagnosis. Specifically, receiver operating characteristic (ROC) curve analyses will be used to determine the relative sensitivity and specificity of OMT for differentiating the two patient groups (PD vs other diagnosis). This provides Area Under the Curve (AUC) which is a metric to evaluate the classifiers output quality.

To examine whether OMT can differentiate PD from age-matched controls, we will use an ANOVA, with group (PD vs control) as a between-subject factor. Again, to adjust for covariates (i.e. disease severity/ cognitive function), we will use analysis of covariance (ANCOVA). Receiver operating characteristic (ROC) curve analyses will also be used here to determine the relative sensitivity and specificity of OMT for differentiating People with PD and healthy controls. This provides AUC which is a metric to evaluate the classifiers output quality.

*Clinical validation aim II. Convergent / Divergent validity*: Pearson's correlations will be used to detail the relationships between OMT measurement and other clinically relevant outcome measures, such as the UPDRS or MoCA.

*Clinical validation aim III. Known intervention response*: To examine whether OMT reduces when in the OFF-medication state in PD, we will use an ANOVA with medication-state (ON vs OFF medication) as repeated measures.

## Discussion

We provide a protocol for the digital measurement of OMT in People with PD using a novel, non-invasive, handheld device. There is no "gold standard" technique for OMT measurement and so the application of OMT measurement protocols varies, consequently limiting generalisability and interpretation of underlying deficits. Investigators who wish to measure and study OMT are left with a choice of various invasive measurement techniques and protocols that differ in many respects [24]. In the process of developing robust and feasible protocols for clinical research it is essential to have evidence-based guidance [50]. This protocol outlines the steps for assessing reliability and clinical validity of OMT using the iTremor ONE device. The primary outcome for the study will be frequency (Hz) of OMT, measured via the handheld iTremor device. Secondary outcomes for the study include cognition, visual function, motor symptoms (UPDRS-III), gait and balance.

### Analytical validation

This protocol was developed in response to evidence for the need for a quick, non-invasive measure of OMT. This protocol aims to assess the reliability of the iTremor ONE device in measuring OMT in People with PD. Previous OMT measurement and methods range from invasive (e.g., piezoelectric techniques) to modern less invasive technological assessment with eye-tracking devices [31, 35, 51–53]. Due to the invasive nature of previous techniques, test retest reliability is performed to measure OMT frequency (Hz) in People with PD. The implications of this are that it is expected to provide the first evidence of the validity and reliability of using a non-invasive, handheld device to measure OMT in People with PD. This study also explores the feasibility of measuring OMT using a hand-held device in a real-world clinical setting. Evidence of an OMT as a biomarker for PD would be hugely beneficial in both supporting clinical judgement and tailoring patient care plans [24].

### Clinical validation

This protocol aims to explore the ability and clinical usefulness of OMT as a potential future biomarker for PD compared to healthy controls, and suspected PD patients for diagnosis and disease monitoring. Early diagnosis would allow for improved patient care plans and the chance to trial neuroprotective therapies to slow down the development and progression of the disease. This protocol also aims to determine the clinical usefulness of OMT measurement in response to medication as a monitor of treatment plans. This understanding again, would allow for overall improved patient care and improved individualised treatments.

## Conclusion

This will be the first study of its kind to non-invasively investigate OMT frequency as a marker/monitor for PD with advanced technology that could be used within the clinic, laboratory, or home. Identifying OMT as a biomarker in PD could better support clinical assessment, enabling improved provision of care to patients with advanced disease monitoring.

## Dissemination policy

On completion of the study, the data will be analysed and tabulated and a Final Study Report prepared. All participating investigators have rights to publish any of the study data, with agreement from the other investigators. Participants will be notified of the outcome of the study via a specifically designed newsletter. Participants can specifically request results which will be provided after the Final Study Report has been compiled.

## Supporting information

**S1 File. SPIRIT checklist.**
(PDF)

## Acknowledgments

This study is based at the Physiotherapy Innovation Laboratory (Website: www.pi-lab.co.uk, Twitter: @Physio_In_Lab).

## Author Contributions

**Conceptualization:** Lisa Graham, Rodrigo Vitorio, Alan Godfrey, Rosie Morris, Samuel Stuart.

**Data curation:** Lisa Graham, Rosie Morris.

**Formal analysis:** Lisa Graham, Rodrigo Vitorio, Samuel Stuart.

**Funding acquisition:** Rodrigo Vitorio, Alan Godfrey, Rosie Morris, Samuel Stuart.

**Investigation:** Lisa Graham, Rodrigo Vitorio, Richard Walker, Alan Godfrey, Rosie Morris, Samuel Stuart.

**Methodology:** Lisa Graham, Rodrigo Vitorio, Alan Godfrey, Rosie Morris, Samuel Stuart.

**Project administration:** Lisa Graham, Rodrigo Vitorio, Alan Godfrey, Rosie Morris, Samuel Stuart.

**Resources:** Samuel Stuart.

**Supervision:** Rodrigo Vitorio, Alan Godfrey, Rosie Morris, Samuel Stuart.

**Validation:** Lisa Graham, Rodrigo Vitorio, Samuel Stuart.

**Writing – original draft:** Lisa Graham, Samuel Stuart.

**Writing – review & editing:** Lisa Graham, Rodrigo Vitorio, Richard Walker, Alan Godfrey, Rosie Morris, Samuel Stuart.

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
