## [Decision Letter · Decision Letter 0]

15 Aug 2024

PONE-D-24-15473Digital measurement of ocular microtremor in Parkinson’s Disease: Protocol to assess reliability and clinical validation.

Dear Dr. Stuart,

Thank you for submitting your manuscript to PLOS ONE. After careful consideration, we feel that it has merit but does not fully meet PLOS ONE’s publication criteria as it currently stands. Therefore, we invite you to submit a revised version of the manuscript that addresses the points raised during the review process.

We look forward to receiving your revised manuscript.

Kind regards,

Karsten Witt

Academic Editor

PLOS ONE

“This study is based at the Physiotherapy Innovation Laboratory (Website: www.pi-lab.co.uk, Twitter: @Physio_In_Lab) and has been funded by a Northumbria University PhD studentship in collaboration with Head Diagnostics Ltd. (Dublin, Ireland). Dr Samuel Stuart is supported, in part, by a Parkinson’s Foundation Post-doctoral Fellowship for Basic  Scientists (PF-FBS-1898-18-21) and a Clinical Research Award (PF-CRA-2073). Private Physiotherapy Educational Foundation funding also supported this research (PPEF – #368; PI: Stuart).”

“This study is based at the Physiotherapy Innovation Laboratory (Website: www.pi-lab.co.uk, Twitter: @Physio_In_Lab) and has been funded by a Northumbria University PhD studentship in collaboration with Head Diagnostics Ltd. (Dublin, Ireland). Dr Samuel Stuart is supported, in part, by a Parkinson’s Foundation Post-doctoral Fellowship for Basic  Scientists (PF-FBS-1898-18-21) and a Clinical Research Award (PF-CRA-2073). Private Physiotherapy Educational Foundation funding also supported this research (PPEF – #368; PI: Stuart).”

“This study is based at the Physiotherapy Innovation Laboratory (Website: www.pi-lab.co.uk, Twitter: @Physio_In_Lab) and has been funded by a Northumbria University PhD studentship in collaboration with Head Diagnostics Ltd. (Dublin, Ireland). Dr Samuel Stuart is supported, in part, by a Parkinson’s Foundation Post-doctoral Fellowship for Basic  Scientists (PF-FBS-1898-18-21) and a Clinical Research Award (PF-CRA-2073). Private Physiotherapy Educational Foundation funding also supported this research (PPEF – #368; PI: Stuart).”

Reviewers' comments:

Reviewer's Responses to Questions

**Comments to the Author**

1. Does the manuscript provide a valid rationale for the proposed study, with clearly identified and justified research questions?

Reviewer #1: Yes

Reviewer #2: Yes

2. Is the protocol technically sound and planned in a manner that will lead to a meaningful outcome and allow testing the stated hypotheses?

Reviewer #1: Yes

Reviewer #2: Partly

3. Is the methodology feasible and described in sufficient detail to allow the work to be replicable?

Reviewer #1: Yes

Reviewer #2: Yes

4. Have the authors described where all data underlying the findings will be made available when the study is complete?

Reviewer #1: Yes

Reviewer #2: No

5. Is the manuscript presented in an intelligible fashion and written in standard English?

Reviewer #1: Yes

Reviewer #2: Yes

6. Review Comments to the Author

You may also provide optional suggestions and comments to authors that they might find helpful in planning their study.

Reviewer #1: This is a very well conceptualized and constructed protocol. All necessary components are included and adequately detailed. This protocol is IRB approved and has been registered on clinicaltrials.gov. I have no reservations in recommending this protocol for publication at this time.

Reviewer #2: • Is the protocol technically sound and planned in a manner that will lead to a meaningful outcome and allow testing of the stated hypotheses?

Yes, with the following caveats:

The protocol indicates that patients will be recruited when they first present at clinic with suspected PD. Those that are confirmed as PD go into the PD group, those that get some other diagnosis (or are healthy) go into the ‘not PD’ group. Superficially this makes good sense, at it allows the researchers to compare two groups at the point of diagnosis. However, the non-PD group will be extremely heterogeneous and may include people with no illness at all, people with non-PD motor disorders and people with other illnesses that could compromise their eye-movements such as stroke or Alzheimer’s disease. I think this could be a potential problem for interpretation, especially if the study finds no group differences.

The justification for the sample size is not very convincing. The purpose of the power analysis is establish the sample necessary to observe a statistically significant difference between the two groups. Given there is prior evidence that OMT differs in PD and controls I would expect to see a formal power analysis to support the sample of 30, irrespective of whether or not the authors wish to draw conclusions about the clinical value of the test.

Why is only the PD group getting a 2nd assessment in the home? It seems necessary to test reliability in both groups in the home to make a valid conclusion about the reliability of iTremor, especially as people homes are much more variable than the lab, so it would not be very surprising to see at least some variation in scores when testing moves from controlled lab environments to homes.

How will usability and acceptability be assessed? The authors mention reporting descriptions of usability and acceptability but not how this date will be collected (e.g. is it based on the researchers reflection? Will patients be asked? will metrics like the time it takes to perform the test be measured?)

Minor comments

Introduction

Line 81-83 : sentence doesn’t make sense. I think because the authors don’t explain what ability is being referred to in line 81.

Line 84: the authors argue that that main problem for using eye-movements to diagnose PD is that some patients don’t understand the instructions on some tasks. They allude to the antisaccade task, but are not explicit. I didn’t quite follow this line of argument, as earlier in the para it is argued that prosaccades are in fact the most best for predicting decline. I would also be surprised if non-dementing PD patients had problems understanding antisaccade or memory guided saccade tasks (at least, no more so than similarly aged controls). The issue of medication is more problematic, but of course if a patient is already on medication isn’t it likely they have been diagnosed with PD, so the use of eye-movement data for diagnosis is no longer necessary? (whereas it it might still be useful for progression monitoring)

Line 304-306: the text suggests patients will be told to abstain from taking their meds, then a week later the team will visit. I assume this not the plan, so this phrasing needs to be tightened up

7. PLOS authors have the option to publish the peer review history of their article (what does this mean?). If published, this will include your full peer review and any attached files.

Reviewer #1: **Yes: **Erin E. Robertson

Reviewer #2: No

---

## [Author Response · Author response to Decision Letter 0]

16 Sep 2024

Please see attached reviewer responses in files section.

---

## [Decision Letter · Decision Letter 1]

24 Oct 2024

Digital measurement of ocular microtremor in Parkinson’s Disease: Protocol for a pilot study to assess reliability and clinical validation.

PONE-D-24-15473R1

Dear Dr. Stuart,

We’re pleased to inform you that your manuscript has been judged scientifically suitable for publication and will be formally accepted for publication once it meets all outstanding technical requirements.

Kind regards,

Karsten Witt

Academic Editor

PLOS ONE

Additional Editor Comments (optional):

Dear Dr. Stuart,

Thank you for the submitted revision of your paper entitled “Digital measurement of ocular microtremor in Parkinson's Disease: Protocol for a pilot study to assess reliability and clinical validation”.

I am pleased to accept the paper in the revised version, I have received the positive evaluation from the reviewers. The review process took some time because I was not able to find suitable reviewers right away, so I apologize for the delay.

Best regards

Karsten Witt

Reviewers' comments:

Reviewer's Responses to Questions

**Comments to the Author**

1. Does the manuscript provide a valid rationale for the proposed study, with clearly identified and justified research questions?

Reviewer #1: Yes

Reviewer #2: Yes

2. Is the protocol technically sound and planned in a manner that will lead to a meaningful outcome and allow testing the stated hypotheses?

Reviewer #1: Yes

Reviewer #2: Yes

3. Is the methodology feasible and described in sufficient detail to allow the work to be replicable?

Reviewer #1: Yes

Reviewer #2: Yes

4. Have the authors described where all data underlying the findings will be made available when the study is complete?

Reviewer #1: Yes

Reviewer #2: Yes

5. Is the manuscript presented in an intelligible fashion and written in standard English?

Reviewer #1: Yes

Reviewer #2: Yes

6. Review Comments to the Author

You may also provide optional suggestions and comments to authors that they might find helpful in planning their study.

Reviewer #1: The revisions these authors have made has resulted in a stronger protocol, more clearly defined protocol. I have no reservation in recommending this manuscript for publication.

Reviewer #2: The authors responses have convincingly addressed the issues raised in my first review. The study should be a worthwhile contribution.

7. PLOS authors have the option to publish the peer review history of their article (what does this mean?). If published, this will include your full peer review and any attached files.

Reviewer #1: No

Reviewer #2: No

---

## [Editor Report · Acceptance letter]

5 Nov 2024

PONE-D-24-15473R1 

PLOS ONE

Dear Dr. Stuart, 

I'm pleased to inform you that your manuscript has been deemed suitable for publication in PLOS ONE. Congratulations! Your manuscript is now being handed over to our production team.

Kind regards, 

on behalf of

Dr. Karsten Witt 

Academic Editor

PLOS ONE